# Long-Term LDL-Apheresis Treatment and Dynamics of Circulating miRNAs in Patients with Severe Familial Hypercholesterolemia

**DOI:** 10.3390/genes14081571

**Published:** 2023-08-01

**Authors:** Dana Dlouha, Milan Blaha, Pavlina Huckova, Vera Lanska, Jaroslav Alois Hubacek, Vladimir Blaha

**Affiliations:** 1Center for Experimental Medicine, Institute for Clinical and Experimental Medicine, 14021 Prague, Czech Republic; hucp@ikem.cz (P.H.); jahb@ikem.cz (J.A.H.); 24th Department of Internal Medicine—Hematology, University Hospital Hradec Králové, 50005 Hradec Králové, Czech Republic; milan.blaha@fnhk.cz; 3Faculty of Medicine in Hradec Králové, Charles University, 50003 Hradec Králové, Czech Republic; blaha@lfhk.cuni.cz; 4Statistical Unit, Institute for Clinical and Experimental Medicine, 14021 Prague, Czech Republic; vela@ikem.cz; 51st Faculty of Medicine, Charles University, 12108 Prague, Czech Republic; 63rd Department of Internal Medicine—Metabolism and Gerontology, University Hospital Hradec Králové, 50005 Hradec Králové, Czech Republic

**Keywords:** miRNA, apheresis, familial hypercholesterolemia, PCSK9 inhibitor, endothelial function, cardiovascular risk

## Abstract

Lipoprotein apheresis (LA) is a therapeutic option for patients with severe hypercholesterolemia who have persistently elevated LDL-C levels despite attempts at drug therapy. MicroRNAs (miRNAs), important posttranscriptional gene regulators, are involved in the pathogenesis of atherosclerosis. Our study aimed to monitor the dynamics of twenty preselected circulating miRNAs in patients under long-term apheresis treatment. Plasma samples from 12 FH patients (men = 50%, age = 55.3 ± 12.2 years; mean LA overall treatment time = 13.1 ± 7.8 years) were collected before each apheresis therapy every sixth month over the course of four years of treatment. Eight complete follow-up (FU) samples were measured in each patient. Dynamic changes in the relative quantity of 6 miRNAs (miR-92a, miR-21, miR-126, miR-122, miR-26a, and miR-185; all *p* < 0.04) during FU were identified. Overall apheresis treatment time influenced circulating miR-146a levels (*p* < 0.04). In LDLR mutation homozygotes (N = 5), compared to heterozygotes (N = 7), we found higher plasma levels of miR-181, miR-126, miR-155, and miR-92a (all *p* < 0.03). Treatment with PCSK9 inhibitors (N = 6) affected the plasma levels of 7 miRNAs (miR-126, miR-122, miR-26a, miR-155, miR-125a, miR-92a, and miR-27a; all *p* < 0.04). Long-term monitoring has shown that LA in patients with severe familial hypercholesterolemia influences plasma circulating miRNAs involved in endothelial dysfunction, cholesterol homeostasis, inflammation, and plaque development. The longer the treatment using LA, the better the miRNA milieu depicting the potential cardiovascular risk.

## 1. Introduction

Familial hypercholesterolemia (FH) is a frequent, severe, and mostly autosomal dominant genetic disorder associated with elevated plasma low-density lipoprotein cholesterol (LDL-C) levels that predisposes patients towards premature atherosclerotic cardiovascular disease, especially if they remain undiagnosed or inadequately treated [1]. The disorder most commonly results from loss-of-function mutations in the low-density lipoprotein cholesterol receptor (*LDLR*) gene encoding LDL receptor protein and genes encoding proteins that interact with the receptor, including apolipoprotein B (APOB), proprotein convertase subtilisin/kexin type 9 (PCSK9), or LDLR adaptor protein 1 (LDLRAP1) [2,3].

Lipoprotein apheresis (LA) is a therapeutic option for patients with severe hypercholesterolemia who have persistently elevated LDL-C levels despite attempts at drug therapy [4]. LA is an extracorporeal elimination technique that removes LDL particles and other pathogenic lipoproteins, such as lipoprotein (a) or triglyceride-rich lipoproteins, from the circulation. The main indications for lipoprotein apheresis are (i) the diagnosis of homozygous FH, (ii) heterozygous FH that is refractory to the standard care and intolerant to routine care, and (iii) patients with lipoprotein (a) and increased resistance to pharmacotherapy [5]. LA is also a potent therapy that impacts inflammation and related mediators [6].

MicroRNAs (miRNAs) are small endogenous noncoding RNAs that regulate the mRNA translation of target genes via the RNA interference pathway, strongly influencing a wide range of cellular processes and biological pathways [7,8]. MiRNAs are released into human plasma and serum to serve as biomarkers for clinical diagnosis, prognosis, and follow-up monitoring of the consequences of treatments [9]. Extracellular miRNAs retain their stability and avoid degradation via incorporation into membrane-derived vesicles or bound to circulating proteins to mediate cell communication with neighbouring or remote cells [10]. Atherosclerosis is characterised by endothelial dysfunction, which promotes inflammatory responses to cholesterol and lipid accumulation within the arterial wall. Endothelial dysfunction encompasses a spectrum of biological processes, and miRNAs have emerged as critical regulators of endothelial gene regulatory networks [11].

Based on our previous study [12], we hypothesised that long-term LA could influence the dynamics of circulating miRNAs involved in atherosclerosis pathophysiology. We have already reported that the long-term elimination of pathogenic lipoproteins from plasma related to endothelial dysfunction and inflammation suggests improving cardiovascular prognosis in most FH patients [13]. Twenty candidate miRNAs were selected based on screening of miRNAs involved in atherosclerosis development (Appendix A). Predominantly, plasma-stable miRNAs targeting genes involved in the regulation of endothelial dysfunction, cholesterol homeostasis, inflammation, and plaque development were measured (Appendix A).

## 2. Materials and Methods

### 2.1. Design and Study Population

We studied a group of 12 patients with familial hypercholesterolemia under long-term apheresis treatment (men = 50%, age = 55.3 ± 12.2 years). The characteristics of the patients have been previously described [14]. DNA-based evidence of a mutation in the *LDLR* gene was the criterion for the diagnosis of homozygous FH (N = 5). None of the patients had a mutation in the *APOB* gene. All patients were treated daily with statins (rosuvastatin 40 mg or 20 mg) combined with ezetimibe (10 mg) or in one subject with fenofibrate (267 mg). At the time of sampling, 6 patients were treated with PCSK9 inhibitors (Table 1).

The patients had been regularly treated with LDL apheresis (immunoadsorption) or rheohemapheresis (cascade filtration) for an average of 13.1 ± 7.8 years. Several events had occurred in some patients during this period: one patient with homozygous FH, who had aortic valve stenosis, underwent a Bentall procedure and triple aortocoronary bypass at the age of 42 years. The second patient with homozygous FH had an ischemic stroke at the age of 64 years. A third homozygous FH patient had a myocardial infarction and aortocoronary bypass surgery at the age of 27 years (five years after starting LA). Plasma collected during the period from 2016 to 2020 was measured in our study. During this time, any cardiovascular events did not occur in any person.

Samples before each apheresis/rheopheresis therapy at 6-month intervals for 4 years of treatment were analysed. Eight complete follow-up (FU) samples were measured in 6 patients, and 7 FU samples were measured in 3 patients. In the remaining three patients, 4 to 6 blood samples were collected during the FU. Characteristics, medical history, laboratory assessments, and medications were noted over the course of the study at the defined FU times.

### 2.2. LDL-Apheresis

Plasma separation was performed using a Cobe-Spectra or Optia continuous centrifugal separator (Terumo, Likewood, CO, USA) in 9 patients. An adsorption–desorption automatic device (Adasorb, Medicap, Germany) controlled repeated fillings and washings of Lipopak adsorbers (Pocard, Moscow, Russia). In 2 patients, Lipocollect adsorbers (Medicollect, Germany) were used. Briefly, patients’ blood was taken via peripheral venous access to a Cobe-Spectra or Optia blood cell separator (Terumo, Likewood, CO, USA) that, acting as a centrifuge, separated plasma and cellular components of the blood. In accordance with the immunoadsorption technique, plasma was pumped through Lipopak affinity columns (Pocard, Moscow, Russia) containing antibodies against the main lipoprotein of LDL-cholesterol—apolipoprotein B100 [15].

### 2.3. Rheohemapheresis

Three patients simultaneously received long-term therapy due to hypercholesterolemia and increased levels of fibrinogen. Rheohemapheresis therapy was used in accordance with Borberg et al. with our own modification [16]. In rheopheresis, plasma is pumped through a filter that separates out lipoproteins and other large molecules. Purified plasma is mixed with blood cells separated earlier and returned back to the patient via another peripheral vein. The adsorption is fully automated; the plasma flow through the adsorption columns is directed by a secondary device, adsorption desorption automat Adasorb (Medicap, Ulrichstein, Germany). To obtain plasma, we used continuous separators (Cobe Spectra or Spectra Optia, Terumo BCT, Lakewood, CO, USA) and Evaflux 4A filters (Kawasumi, Tokyo, Japan) to wash the obtained plasma. The flow through the filter was controlled using a CF100 automatic machine (Infomed, Geneva, Switzerland). Anticoagulation was performed using a combination of heparin and ACD-A (Baxter, Munich, Germany). Of the circulating plasma volume, 1 to 1.5% was washed and was calculated via the on-board computer of the blood cell separator. The procedures were performed from blood taken from the peripheral vein in the elbow pit or in the forearm.

### 2.4. Plasma Samples

Blood samples (10 mL) were collected in EDTA-containing tubes and centrifuged at 1500× *g* for 15 min at room temperature. Plasma samples were processed within 30 min of blood collection, aliquoted into RNase-free tubes, and stored at −80 °C before RNA extraction. The median storage time (ST) of plasma samples was 966 days (range: 10 to 1882 days). Free haemoglobin levels (f-Hb), one of the pre-analytical factors that significantly influence circulating plasma miRNA levels, were measured as an indicator of haemolysis. The concentration of f-Hb was calculated as we reported previously [12]. Samples with f-Hb above 25 mg/dL (N = 8) were considered to be affected by haemolysis and were excluded from the analysis.

Total serum cholesterol (TC), low-density lipoprotein cholesterol (LDL-C), high-density lipoprotein cholesterol (HD-CL), and triglycerides (TAG) were determined using a commercial kit on a Modular Roche analyser according to the manufacturer’s instructions [17].

### 2.5. MiRNA Measurement

Total RNA was extracted from 200 μL plasma using the miRCURY^TM^ RNA isolation kit for biofluids (Exiqon, Vedbaek, Denmark) as we reported previously [12,18]. SYBR green-based real-time quantitative PCR was performed using a QuantStudio6 instrument (Thermo Scientific, Waltham, MA, USA). Passive reference dye (ROX^TM^ 50 nm) was included for all PCRs. Interplate calibrators and spike-in controls were included in each analysis to ensure the quality of RNA isolation, cDNA synthesis, and PCR. Twenty-three individual miRCURY miRNA assays (Qiagen, Hilden, Germany) were used for quantitative PCR (Appendix A). As endogenous miRNA controls, hsa-miR-103-3p, hsa-miR-191-5p and hsa-let-7a-5p were selected. GenEx SW (Multid Analysis AB, Göteborg, Sweden) was used for miRNA expression analysis. Four miRNAs, namely, miR-758, miR-370, miR-33a and miR-34a, with a call rate <50% (i.e., more than 50% of the data were invalid for that miRNA) were excluded from the analysis. Ct values higher than 35, which indicated a low concentration of miRNAs in plasma, were replaced by the value 35. The Genorm and NormFinder algorithms selected let-7a-5p for normalization. The missing data, very low miRNA levels, were replaced by deltaCt+2 (which represents at least 1/4 of the detectable miRNA amount). Data were converted to relative quantities and to the log transformations of the values.

### 2.6. Statistical Analysis

Data were tested for normality using the Shapiro–Wilk test and are presented as the mean ± SD or as median (IQR). In the graphs, the mean and standard error are shown. The nonparametric Kruskal–Wallis test was used for biochemical parameter comparisons. Partial correlation was used for comparison of biochemical parameters with miRNA quantity, and then Bonferroni correction was applied to significance levels. GenEx SW (Multid Analysis AB, Göteborg, Sweden) was used for miRNA expression analysis. Data were converted to relative quantities and to the log transformations of the values. Calculations were performed using JMP software 16.2.0 2020–2021 SAS Institute Inc. (Cary, NC, USA). A *p* value <0.05 was considered statistically significant. MicroRNA gene targets were predicted using the miRWalk program released January 2022.

## 3. Results

### 3.1. Dynamics of miRNA Quantity during FU

The overall apheresis treatment time ranged between 0 and 22.2 years in our study group. There was no emergence within the monitored period of new coronary events or deaths in any patient treated with LDL-apheresis or rheopheresis.

Only one patient (LDLR homozygote, female, 54 years) initiated her first-ever apheresis in the study. The patient was treated daily with rosuvastatin (40 mg) in combination with ezetimibe (10 mg) and anopyrine (100 mg). The sample collected before the first LA ever we used as a baseline for comparison with other measured plasma samples. Relative quantification was performed to this sample. Since patients were treated with LA for a long time, the sample from an untreated subject was ideal for comparing of treatment effect.

Sixteen miRNAs were successfully measured, namely: miR-126-3p; miR-155-5p; miR-122-5p; miR-125a-5p; miR-146a-5p; miR-181a-5p; miR-17-5p; miR-144-5p; miR-26a-5p; miR-29c-3p; miR-143-5p; miR-92a-3p; miR-27a-3p; miR-365a-3p; miR-185-5p; and miR-21-5p. As we reported previously [18], there exists a correlation between the quantity of miRNAs in plasma and ST. After ST adjustment, we found changes in the quantity of miR-92a (*p* < 0.0001), miR-185 (*p* < 0.01), miR-21 (*p* < 0.02), miR-126 and miR-122 (all *p* < 0.03), and miR-26a (*p* < 0.04) at the FU time points. Most miRNAs oscillated during FU; only miR-92a showed an increasing trend (Figure 1).

We divided patients into 2 groups, with mean LA overall treatment times of 17.9 ± 4.3 years (N = 6) and 5.5 ± 3.3 years (N = 6), and compare effect of treatment duration. In patients with shorter treatment times, we found a higher relative quantity of plasma miR-146, briefly 0.52 ± 0.04 vs. 0.39 ± 0.03; *p* < 0.04 (Appendix A). We didn’t find differences in separation methods on plasma miRNA levels (Appendix A).

Four patients had type 2 diabetes mellitus (T2DM). Further statistical analyses revealed lower levels of miR-155 in patients with T2DM during FU (*p* = 0.01; Appendix A).

### 3.2. MiRNA Quantity and LDLR Mutation Genotype

In *LDLR* mutation homozygotes (N = 5; 42% of patients), we detected higher plasma levels of miR-155 (*p* < 0.0006), miR-126 (*p* < 0.01) and miR-181 and miR-92a (both *p* < 0.03) compared to *LDLR* heterozygotes (N = 7; 58%) (Figure 2A). Moreover, we identified significant differences in miR-155 quantity between *LDLR* homo vs. heterozygous patients during FU (*p* < 0.05) (Figure 2B).

### 3.3. PCSK9 Inhibitor Treatment and miRNA Levels

Six patients (4 *LDLR* homozygotes/2 *LDLR* heterozygotes) were under combined LA treatment with PCSK9 inhibitors. Two patients were treated with PCSK9 inhibitors throughout the FU. Three patients started treatment with PCSK9 inhibitors during the FU, and one patient had interrupted treatment at 2 FU points. Treatment with PCSK9 inhibitors affected the plasma levels of seven miRNAs. Briefly, increased quantities of miR-126, miR-27a, miR-26a (*p* < 0.002), miR-122 and miR-155 (*p* < 0.02) and miR-92a (*p* < 0.04) were determined in patients treated with PCSK9 inhibitors (Figure 3).

### 3.4. Correlation between miRNAs and Biochemical Parameters

Levels of measured biochemical and lipidemic parameters, including TC, LDL-C, HDL-C, and TAG, were stable during FU (Appendix A). We detected variability only between *LDLR* genotypes, with slightly higher levels of plasma LDL-C in *LDLR* homozygotes (*p* < 0.03). In contrast, the levels of TAG and glycaemia were lower in *LDLR* homozygotes than in heterozygotes (*p* < 0.02 and *p* < 0.05, Appendix A). PCSK9 inhibitor treatment did not affect any parameters.

Partial correlation adjusted for ST revealed all miRNAs were associated with measured biochemical parameters (*p* < 0.05). After Bonferroni correction, we identified significance (*p* < 0.0004) in eight of sixteen miRNAs. A negative relationship between the relative quantity of miR-92a, miR-181a, and miR-185 and plasma levels of ApoB (all *p* < 0.0004) was detected. Furthermore, an inverse correlation was found between miR-27a and Lp(a) (*p* < 0.0003). In contrast, a positive correlation was detected between miR-155 and TAG (*p* < 0.0001; Table 2). MiR-122, miR-21 and miR-365a were inversely correlated with ALT (*p* < 0.0001), and miR-122 also with AST (*p* < 0.0001; Table 2).

## 4. Discussion

Our centre is the first one in the Czech Republic to start the treatment of FH patients using LDL apheresis/rheopheresis. Over 38 years of performing LDL apheresis, we encountered 19 patients who were either homozygous or more severely heterozygous and required lipid apheresis. The aim of the current study was to monitor the dynamics of twenty preselected circulating miRNAs in patients under long-term apheresis treatment. To the best of our knowledge, no work has been published addressing the issue of miRNAs in a larger cohort of patients followed for such a long period of time for familial hypercholesterolemia.

### 4.1. Long-Term LA and Changes in miRNA Quantity

Our retrospective study detected quantitative changes in miRNAs predominantly associated with endothelial dysfunction (miR-92a and miR-126), cholesterol metabolism (miR-122, mir-26, and miR-185), and vascular remodelling (miR-21) in FH patients under long-term apheresis treatment. The majority of miRNAs oscillated during FU points, and only miR-92a showed an increasing trend. MiR-92a, a member of the miR-17-92 cluster, is important for cell angiogenesis, apoptosis, migration, and inflammation [19,20,21]. Endothelial miR-92a is responsive to atheroprone micro-environmental cues, and its expression level is increased by oscillatory low shear stress and oxidised LDL [10,22]. Among the targets of miR-92a are sirtuin 1 (*SIRT1*), Krüppel-like factor 2 (*KLF2*), and *KLF4*, all of which positively regulate eNOS-derived NO. By inhibiting these endothelial-protective molecules, miR-92a can promote the endothelial innate immune response and the ensuing vascular inflammation [22]. Recently, it has been demonstrated that extracellular miR-92a mediates endothelial cell-macrophage communication. MiR-92a can be transported to macrophages via extracellular vesicles to regulate KLF4 levels, thus leading to atheroprone phenotypes of macrophages and hence atherosclerotic lesion formation [10]. A strong association between circulating miR-92a levels and coronary artery disease in humans was previously reported [23]. The increased quantity of circulating miR-92a during FU probably reflects worsened endothelial dysfunction in FH patients despite long-term LA treatment combined with hypolipidemic drugs of the new generation. Despite an intensive hypolipidemic intervention, patients with severe hypercholesterolemia are not reaching the desired LDL-C targets (this is evident in Appendix A, which shows biomarkers of FH patients during aphaeresis treatment). Long-term monitoring of miRNAs in patients with FH could elucidate the progression of atherosclerosis.

Detection of higher levels of miR-146 in patients with shorter LA treatment times could support the finding that FH patients benefit from long-term apheresis procedures that reduce inflammation, which might be important for the prevention of cardiovascular events [5].

The plasma concentration of miR-155 was lower in subjects with diabetes during FUs. Decreased levels of miR-155 probably reflect declined pancreatic function and insulin resistance as reported previously [24].

### 4.2. LDLR Mutation

In *LDLR* mutation homozygotes compared to heterozygotes, we identified higher levels of miRNAs predominantly involved in vascular integrity, angiogenesis and inflammation (miR-126, miR-92a, miR-155 and miR-181a). MiR-155 was the only miRNA that increased during whole FUs. MiRNA-155 is a typical multifunctional miRNA that has been associated with the occurrence and development of atherosclerosis by regulating the functions of CD4+ T lymphocytes, monocytes/macrophages, endothelial cells (ECs), and vascular smooth muscle cells [25,26]. MiR-155 is upregulated in human cardiac disease [27] and is significantly increased in plasma and plaques in atherosclerotic patients [28]. Higher levels of miR-155 may reflect worse vascular integrity in *LDLR* homozygous patients.

### 4.3. PCSK9 Inhibitor Treatment

Finally, add-on treatment with PCSK9 inhibitors showed increased plasma levels of miRNAs participating in cholesterol metabolism (miR-122 and miR-27a), endothelial dysfunction (miR-126 and mir-26a), and inflammation (miR-92a, miR-155, and miR-125a).

MiR-27 was the most increased miRNA in PCSK9 inhibitor-treated patients. Several studies have identified important roles for miR-27 in lipid metabolism, inflammation, angiogenesis, adipogenesis, oxidative stress, the renin-angiotensin system, insulin resistance, and type 2 diabetes, which have important roles in the onset and outcome of atherosclerosis [29,30]. Interestingly, miR-27a also directly decreases *LRP6* and *LDLRAP1*, two other key players in the LDLR pathway that are required for efficient endocytosis of the LDLR-LDL-C complex in the liver [31]. Moreover, serum levels of miR-27a were found to be inversely related to two cholesterol transporters: *ABCA1* and *ABCG1* gene expression in coronary heart disease patients [32]. It should be noted that four patients under combined LA treatment with PCSK9 inhibitors were *LDLR* homozygotes. Three up-regulated miRNAs: miR-92a, miR-155, and miR-126 were also increased when subjects with different *LDLR* genotypes were compared. In such a small number of analysed individuals, a strong effect of *LDLR* homozygotes on the result cannot be excluded. In addition, we cannot exclude the distortion of the results due to differences in the PCSK9 inhibitors usage protocol.

### 4.4. Biochemical Parameters

Unlike in our previous studies [12,14], we only analysed samples before each LA in approximately 6-month intervals in the present study. Stable levels of lipidemic parameters, including TC, LDL-C, HDL-C, and TAG, during FU probably reflect the persistent effect of LA treatment. The lower LDL-C at the last FU interval may reflect the effect of combined therapy using PCSK9 inhibitors that were used from that time point. Not surprisingly, LDL-C concentration was higher in LDLR homozygotes, but interestingly, lower levels of glycemia and TAG were detected in these patients.

Furthermore, we identified an inverse correlation between the levels of ApoB and miR-92a, miR-185, and miR-181a. As we mentioned above, miR-92a targets *SIRT1,* which associates with many modulators regulating lipid metabolism and results in increased expression of sterol regulatory element-binding proteins (SREBPs), which acts as a key modulator in lipid synthesis [33]. Recently it has been demonstrated that miR-92a promoted HUVEC apoptosis and suppressed proliferation of ox-LDL-induced HUVECs by targeting *SIRT6* expression and activating MAPK signalling pathway [34]. MiR-185, a liver-specific miRNA, targets sterol regulatory element-binding transcription factor (*SREBF1*), which encodes SREBP1. SREBP1 plays a crucial role in regulating cholesterol, and fatty acid metabolism induces the expression of genes involved in de novo cholesterol biosynthesis and LDL uptake [35]. Moreover, *SREBF1* targets, i.e., the genes *LDLR*, *FASN,* and *INSIG,* are involved in cholesterol and fatty acid metabolism (https://rgd.mcw.edu/wg/home/pathway2/molecular-pathways2/srebf-targets/ (accessed on 22 January 2022)). An in vitro study on human hepatoma cells showed that miR-185 controls cholesterol uptake by directly targeting *LDLR* and the LDLR-destabilizing RNA-binding protein KH-type splicing regulatory protein (KSRP), adding another layer of complexity to the mechanism by which miR-185 regulates cholesterol homeostasis [36]. MiR-181a is another predominant atheroprotective miRNA. MiR-181a targets phosphatase and tensin homolog (*PTEN*). Loss of PTEN activates PI3K/AKT/mTOR pathway, which in turn upregulates SREBP and LDLR. LDL is then hydrolysed to free fatty acids and free cholesterol in lysosome [37]. Generally, both miR-181-5p and miR-181-3p reduce the expression of genes involved in inflammation, such as adhesion molecules and inhibitors of the inflammatory signalling pathway. Both miRNAs also suppress the recruitment of macrophages into lesions [38]. Recently, another group showed that miR-181-5p and miR-181-3p cooperatively inhibit NF-κB signalling by binding to TGF-β-activated kinase 1-binding protein (TAB2) and NF-κB essential modulator (NEMO), respectively [39].

Furthermore, we detected an inverse correlation between Lp(a) and miR-27a levels. It was reported miR-27a decreases low-density lipoprotein receptor-related protein 6 (LRP6) and low-density lipoprotein receptor adapter protein 1 (LDLRAP1), two other key players in the LDLR pathway [30]. MiR-27a/b affects the efflux, influx, esterification and hydrolysis of cellular cholesterol by regulating the expression of *ABCA1*, *APOA1*, *LPL*, *CD36* and *ACAT1* [40].

Not surprisingly, expression of miR-155, a regulator of cholesterol and fatty acid metabolism, was positively associated with TAG concentration in our study Wang et al. [41] reported that liver X receptor (*LXR)α* is the target gene of miR-155, and silencing miR-155 reduced the expression of *SREBP1* and *FAS.* An in vivo study showed that upregulation of miR-155 decreased hepatic lipid accumulation mainly by suppressing the LXRα-dependent lipogenic signalling pathway. All mentioned miRNA targets act in regulating pathways of cholesterol and fatty acid metabolism. The inverse correlation between ApoB and miR-92a, miR-181a, and miR-185a, as well as between Lp(a) and miR-27a, suggests an indirect atheroprotective role of analysed miRNAs. Positive correlation between plasma miR-155 and TAG may provide a new biomarker for clinical usage.

The changes in circulating miRNA levels relate to the pathophysiological background of dyslipidaemia and/or the specific type of LDL receptor mutation (homozygote or heterozygote patients) in familial hypercholesterolemia. Our results could help to elucidate the pathways critical for modulation of lipid metabolism, enhancement of endothelial function, inhibition of inflammation, improvement of plaque stability, and immune regulation and thus provide new avenues for tailored therapies.

The first limitation of this single-centre retrospective study is that include the relatively small number of patients, which may affect the accuracy of our results. Moreover, thanks to the advancement of science and the development of new effective hypolipidemics, especially PCSK9 inhibitors, some patients are achieving target LDL cholesterol levels without the need for lipid apheresis support. Thus, our findings need to be verified by multicentre prospective studies. Furthermore, since the lipoprotein apheresis treatment technique is carried out only in a few large medical centres, many FH patients are unable to receive apheresis treatment, resulting in a particular bias in patient selection

Second, our study group is not completely homogeneous, which may have influenced the occurrence or dynamics of miRNAs. There exist differences in overall treatment time between patients. The cohort is not completely identical in some other parameters either. However, we have performed basic clinical tests at the same time, such as mineral levels, indicators of proper liver and kidney function, endothelial activity, coagulation parameters, and others. Changes in these parameters were correlated with changes in miRNA levels and dynamics to be as close as possible to the clinical condition and also to assess whether changes in some clinical parameters are related to miRNA levels.

Third, two separation methods (immunoadsorption vs. cascade filtration) were used in our study. MiRNAs are released into circulation bound to different types of vesicles, such as apoptotic bodies, microvesicles, exosomes, and lipoproteins. We did not expect to find differences between apheresis and rheopheresis because both methods remove lipoproteins from plasma.

## 5. Conclusions

Long-term monitoring has shown that LDL apheresis in patients with severe familial hypercholesterolemia has an impact on plasma-circulating miRNAs involved in endothelial dysfunction, cholesterol homeostasis, inflammation, and plaque development. The longer the treatment using LDL apheresis, the better the miRNA milieu depicting the potential cardiovascular risk.

## Figures and Tables

**Figure 1 genes-14-01571-f001:**
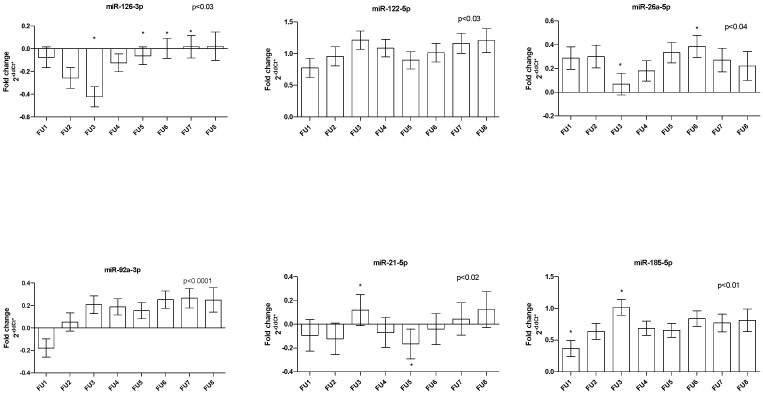
Dynamics of circulating miRNAs in patients under long-term LA treatment during FU. Bar graphs demonstrate the mean ± standard error. * depicts differences between FUs.

**Figure 2 genes-14-01571-f002:**
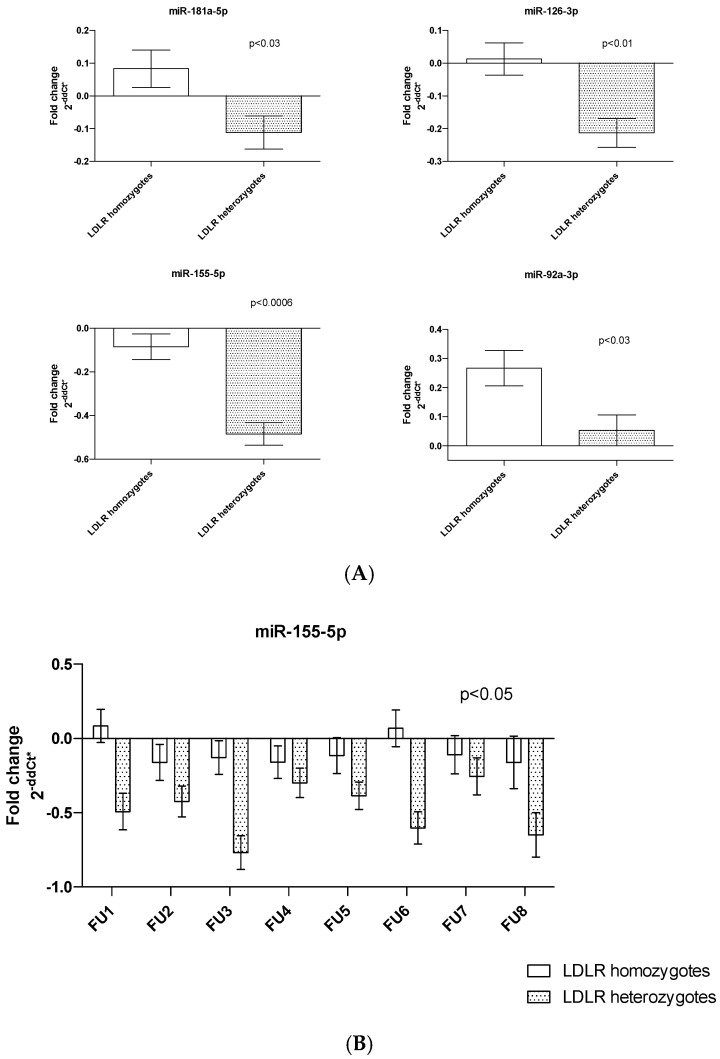
Differences in plasma miRNA quantity (**A**) in *LDLR* mutation carriers of one or two risk alleles; (**B**) changes in the quantity of miR-155 levels according to *LDLR* genotype during FU. Bar graphs demonstrate the mean ± standard error.

**Figure 3 genes-14-01571-f003:**
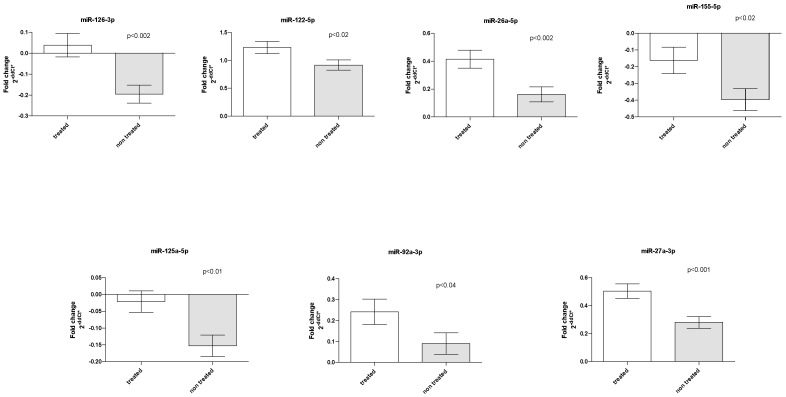
PCSK9 inhibitor treatment effect on miRNA levels. Bar graphs demonstrate the mean ± standard error.

**Table 1 genes-14-01571-t001:** Baseline characteristics of patients.

Male/female (N)	6/6
Age (years)	55.3 ± 12.2
BMI (kg/m^2^)	28.1 ± 4.3
Current smokers and ex-smokers/non-smokers	5/12
Hypertension	7
Diabetes mellitus	4
LDLR mutations: (N) homozygotes/heterozygotes	5/7
Duration of apheresis treatment (years)	13.1 ± 7.8
Hypolipidemic treatment	>13 years
ACE-i/ARB	5
Beta-blockers	8
Antithrombotic drugs	10
PCSK9 inhibitors	6 (2 patients: alirocumab, 150mg per 14 days; 4 patients: evolocumab; 2 subjects with starting dose of 420 mg per 14 days; then all: 140 mg per 14 days)
Statins	12 (all: rosuvastatin 40 mg or 20 mg + ezetimibe 10 mg)

Data are expressed as the mean ± SD or factor proportion. ACE-I—angiotensin-converting-enzyme inhibitors; ARB—angiotensin II receptor blockers; BMI—body mass index; LDLR—low density lipoprotein receptor; PCSK9—proprotein convertase subtilisin/kexin type 9; SD—standard deviation.

**Table 2 genes-14-01571-t002:** Nonparametric Spearman’s correlation of miRNAs and biochemical parameters. A *p* value <0.0004 was considered statistically significant after Bonferroni correction.

	miR-181a		miR-17		miR-122		miR-126		miR-144		miR-26a		miR-155		miR-125a	
	ρ	*p*	ρ	*p*	ρ	*p*	ρ	*p*	ρ	*p*	ρ	*p*	ρ	*p*	ρ	*p*
TC	0.3615	0.0014	−0.2123		−0.0144		−0.0749		−0.1559		−0.2071		−0.2297	0.05	−0.2443	0.03
LDL-C	−0.3736	0.0010	−0.2577	0.03	−0.0761		−0.2639	0.02	−0.2065		−0.2802	0.01	−0.2825	0.01	−0.242	0.04
HDL-C	−0.0045		0.0701		0.0244		0.329	0.004	0.1079		0.0237		−0.0783		−0.1706	
TAG	0.1861		0.1213		0.207		0.2049		0.1503		0.279	0.02	0.4646	**<0.0001**	0.2734	0.02
Glycemia	0.2729	0.018	0.2476	0.03	0.1385		0.2329	0.04	0.0412		0.2614	0.02	0.3197	0.005	0.2397	0.04
ApoB	−0.4345	**<0.0001**	−0.3899	0.0005	−0.0483		−0.0798		−0.3069	0.007	−0.2787	0.02	−0.1699		−0.2476	0.03
Lp(a)	−0.1981		−0.0329		−0.0824		−0.2521	0.03	−0.1721		−0.3757	0.0009	−0.402	0.0004	−0.3277	0.004
ALT	0.0449		−0.0837		−0.6535	**<0.0001**	−0.0254		0.1982		−0.1937		−0.0994		−0.1	
AST	0.0704		−0.0409		−0.5947	**<0.0001**	−0.0547		0.1686		−0.1758		−0.1239		−0.0631	
Creatinine	0.08		0.0953		−0.2559	0.03	0.0285		0.1406		0.021		−0.0004		−0.0532	
	**miR-29c**		**miR-143**		**miR-92a**		**miR-27a**		**miR-146a**		**miR-365a**		**miR-21**		**miR185**	
	**ρ**	** *p* **	**ρ**	** *p* **	**ρ**	** *p* **	**ρ**	** *p* **	**ρ**	** *p* **	**ρ**	** *p* **	**ρ**	** *p* **	**ρ**	** *p* **
TC	−0.2294	0.05	−0.0456		−0.2245		−0.234	0.04	−0.206		−0.2236		−0.1017		−0.2411	0.04
LDL-C	0.0743		−0.0946		−0.2737	0.02	−0.3157	0.006	−0.1969		−0.2722	0.02	−0.0617		−0.2328	0.04
HDL-C	−0.0859		0.0522		0.0519		−0.0085		−0.0712		0.095		−0.1597		0.0252	
TAG	0.1098		0.2533	0.03	0.1418		0.3316	0.004	0.207		−0.0113		0.2645	0.02	−0.021	
Glycemia	−0.3146	0.006	0.2684	0.02	0.2539	0.03	−0.3095	0.007	0.2016		−0.1855		0.3846	0.0007	0.0454	
ApoB	−0.3138	0.006	−0.1161		−0.4947	**<0.0001**	−0.2815	0.01	−0.3562	0.002	−0.2685	0.02	−0.0771		−0.4029	**0.0003**
Lp(a)	−0.072		−0.1971		−0.1433		−0.4111	**0.0002**	−0.1347		0.0959		−0.1523		0.0754	
ALT	−0.0347		0.2013		−0.1648		−0.3581	0.002	0.0594		−0.5398	**<0.0001**	0.4497	**<0.0001**	0.0673	
AST	0.0535		0.1607		−0.1481		−0.2794	0.02	0.0806		−0.3046	0.008	0.2647	0.02	0.1144	
Creatinine	0.1344		0.2248		0.062		−0.064		−0.0518		0.0058		−0.004		0.1229	

## Data Availability

Data is contained within the article or Appendix A.

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
