# Peer review of "Long-Term LDL-Apheresis Treatment and Dynamics of Circulating miRNAs in Patients with Severe Familial Hypercholesterolemia"

_genes, 2023, doi:10.3390/genes14081571_

Round 1

Reviewer 1 Report

The authors focused on the evaluation of the dynamics of twenty preselected circulating miRNAs in patients with familial hypercholesterolemia under long-term apheresis treatment. 

1- The authors used the one patient (LDLR homozygote) as the baseline for comparison with other measured plasma samples, this case should be more detail in clarification. 

2- Observations of the correlation between miRNAs and biochemical parameters were performed; only eight miRNAs were presented; others were not. So authors should give more information about them. Additionally, The correlations between miRNAs and biochemical indicators are scattered. It meant that clinical usage of these miRNAs was difficult. Authors should combine meaningful mirnas together for analysis to increase their association with clinical indicators.

Authors should pay more force on editing the grammar. A lots of sentences are not clear and well-structure. 

Author Response

Dear Reviewer,

We appreciate your comments very much and we used them for the improvement of our manuscript. Our answers and explanations of the individual points in detail, please, see in the accompanying text.

In the original text, changes are highlighted.

Reviewer 2 Report

The authors aimed to monitor the dynamics of twenty preselected circulating miRNAs in patients under long-term apheresis treatment. Plasma samples from 12 FH 20 patients were collected before each apheresis therapy every 6th month over the course of 4 years of treatment. Eight complete follow-up (FU) samples were measured in each patient. Dynamic changes in the relative quantity of 6 miRNAs (miR-92a, miR-21, miR-126, miR-122, miR-26a, and miR-185; all p<0.04) during FU were identified. Overall apheresis treatment time influenced circulating miR-146a levels. In LDLR mutation homozygotes (N=5) compared to heterozygotes, we found higher plasma levels of miR-181, miR-126, miR-155, and miR-92a. Treatment with PCSK9 inhibitors (N=6) affected the plasma levels of 7 miRNAs. They concluded that LA in patients with severe FH influences plasma circulating miRNAs involved in endothelial dysfunction, cholesterol homeostasis, inflammation, and plaque development. The longer the treatment using LA, the better the miRNA milieu depicting the potential cardiovascular risk.

The topic is original and interesting. The study is well designed and nicely presented, although I cannot see the figures in the pdf version.

Comments:

1.       Figures are not inserted into the text!!!

2.       Some other keywords could be added including: PCSK9 inhibitor; endothelial function; cardiovascular risk.

3.       Line 48, 79, 209 and 239: font size should be corrected.

4.       The differences in absorption techniques that used might influence the effect of LA on miRNa levels. This should be discussed and mentioned as a limitation.

5.       Why did FH patients not receive inclisiran?

6.       Was there any difference between diabetic and non-diabetic patients in miRNA levels?

7.       Mean LA overall treatment time was 13.1±7.8 years. During these years, how many cardiovascular events occurred?

English needs editing.

Author Response

(The authors gave the same response as above.)

Reviewer 3 Report

In this manuscript, Dana Dlouha and colleagues, studied the expression of twenty miRNA in patients under long- term apheresis treatment. They showed through long term monitoring, that, LA in patients with familary hypercholesterolemia influences plasma circulating miRNAs involved in several disease such as endothelial dysfunction, inflammation and plaque development. The manuscript needs to be re-shaped for a better understanding of the content. It is hard for the reader to understand the goal of the study. The aim of the study must be clear throughout the study and supported by the data. The discussion does not strongly support the results. I recommend making it more fluid and not dividing it into paragraphs. Furthermore, the study's limitations need to be discussed in more detail.  However, this manuscript needs major revisions before it can be considered suitable for publication in Genes.

Please, find below the major concerns:

-    The authors are requested to explain in detail in the text the choice of the 20 miRNAs analysed.

-     A limitation of this study is that few people were recruited for this study. It is advisable to increase the number of samples to obtain more statistical data.

-       The authors should detail why only one subject was treated with statins and fenofibrate  (267 mg).

-          It must also be specified why 6 patients were treated with PCSK9 inhibitors.

-          The caption of table 1 is shown twice.

-          There is a lot of confusion in the design and study of the population, the authors need to rewrite more fluidly from line 88 to line 91.

-          The authors forgot to add figures 1, 2 and 3 in the results section. It is impossible to review this part without the figures.

-          LDLR mutation homozygotes (N=5), Heterozygotes are ?

-          I would strongly recommend a deep revision of this manuscript to enrich the quality of the paper.

Minor editing of English language required

Author Response

(The authors gave the same response as above.)

Round 2

Reviewer 2 Report

There is a significant improvement. 

Only minor editing is needed. 

Reviewer 3 Report

The authors answered almost all the questions put to them and the quality of the work has improved considerably. I consider the work suitable for publication.

Minor editing of English language required